# Photocatalytic Reduction of CO$_2$ to Methanol Using a Copper-Zirconia Imidazolate Framework

**Sonam Goyal** [1,*], **Maizatul Shima Shaharun** [1], **Ganaga Suriya Jayabal** [1], **Chong Fai Kait** [1], **Bawadi Abdullah** [2] and **Lim Jun Wei** [1]

[1] Department of Fundamental and Applied Sciences, Universiti Teknologi PETRONAS, Seri Iskandar 32610, Perak, Malaysia; maizats@utp.edu.my (M.S.S.); ganagasuriya97@gmail.com (G.S.J.); chongfaikait@utp.edu.my (C.F.K.); junwei.lim@utp.edu.my (L.J.W.)

[2] Chemical Engineering Department, Universiti Teknologi PETRONAS, Seri Iskandar 32610, Perak, Malaysia; bawadi_abdullah@utp.edu.my

\* Correspondence: sonam_g03236@utp.edu.my

**Abstract:** A set of novel photocatalysts, i.e., copper-zirconia imidazolate (CuZrIm) frameworks, were synthesized using different zirconia molar ratios (i.e., 0.5, 1, and 1.5 mmol). The photoreduction process of CO$_2$ to methanol in a continuous-flow stirred photoreactor at pressure and temperature of 1 atm and 25 °C, respectively, was studied. The physicochemical properties of the synthesized catalysts were studied using X-ray diffraction (XRD), X-ray photoelectron spectroscopy (XPS), and photoluminescence (PL) spectroscopy. The highest methanol activity of 818.59 µmol/L.g was recorded when the CuZrIm1 catalyst with Cu/Zr/Im/NH$_4$OH molar ratio of 2:1:4:2 (mmol/mmol/mmol/M) was employed. The enhanced yield is attributed to the presence of Cu$^{2+}$ oxidation state and the uniformly dispersed active metals. The response surface methodology (RSM) was used to optimize the reaction parameters. The predicted results agreed well with the experimental ones with the correlation coefficient ($R^2$) of 0.99. The optimization results showed that the highest methanol activity of 1054 µmol/L.g was recorded when the optimum parameters were employed, i.e., stirring rate (540 rpm), intensity of light (275 W/m$^2$) and photocatalyst loading (1.3 g/L). The redox potential value for the CuZrIm1 shows that the reduction potential is −1.70 V and the oxidation potential is +1.28 V for the photoreduction of CO$_2$ to methanol. The current work has established the potential utilization of the imidazolate framework as catalyst support for the photoreduction of CO$_2$ to methanol.

**Keywords:** methanol; optimization; photocatalytic reduction; imidazolate framework

## 1. Introduction

New inventions are targeted to be low-cost and eco-friendly, which are commonly seen in the carbon industry nowadays. The artificial photosynthesis that converts CO$_2$ and H$_2$O to valuable energy-bearing compounds, for example, methanol is one of the most alluring methods to address the energy crisis and global warming issues [1]. The conventional productivity of methanol from CO$_2$ transformation under light irradiation is very low due to the activation barrier and the photo-corrosion process. Since the last decade, different types of catalysts and physical methods have been developed to alleviate the greenhouse effect [2]. In fact, photoreduction of CO$_2$ to methanol is now getting more popular than hydrogenation. By employing photocatalytic CO$_2$ conversion, sustainable fuel can be generated and more value-added chemicals can be acquired from the waste CO$_2$. This method is one of the prominent ways in CO$_2$ utilization as it is relatively simple and low-cost. Water and solar light are the common renewable sources in photocatalytic CO$_2$ conversion.

In recent years, metal-organic frameworks (MOFs) have been reported as a potential catalyst for photocatalysis. The new generation of catalyst known as porous MOFs [3] has

been widely investigated since the 1990s. MOFs are self-assembled by the coordination of metal cations/clusters with organics linkers. The MOFs can be used in several applications such as nonlinear optics, molecular recognition, sensors, catalysis, gas storage and separation process [4]. Furthermore, MOFs have been employed in dye's degradation [5], water splitting [6], and photocatalytic $CO_2$ reduction or $H_2$ production [7]. Most MOFs have linkers based on imidazolates or aromatic carboxylates. When an imidazolate is employed as a linker, the resulting material is often referred to as a Zeolitic Imidazolate Framework (ZIF), as the resultant crystal structures are quite similar to those of zeolites [8,9]. ZIFs have been employed in some applications related to permeable structure with remarkable thermal and chemical stabilities.

Different metal ions such as $Cu^{2+}$ [10–12], $Zn^{2+}$ [8,9], $Zr^{4+}$ [13–15], $Fe^{2+}$ [16] $Ti^{4+}$ [17], $Co^{2+}$ [18] and $Cd^{2+}$ [19] can be found with ZIFs. Most ZIFs are unstable at high temperature, making them unsuitable for high-temperature applications although they are mechanically robust. Nevertheless, ZIF is a potential catalyst for the photo-driven reaction. As compared to the single metal-doped imidazolate framework, combining two metals in the imidazolate support can improve the photocatalytic performance of imidazolate. Previous studies showed that Cu/ZnO@MOF-5 [20], Cu-porphyrin-based MOF [21], and CuTPP-based MOF [1] performed better than the single metal-doped MOF in photocatalytic $CO_2$ reduction.

From the open works of literature, the effect of reaction parameters on the methanol production is seldom reported. Practically speaking, there is a requirement for developing an appropriate design method for photocatalysis process. The Design of Experiment (DOE) determines the combination of numerous input factors that would affect the methanol productivity. By manipulating multiple inputs at the same time, DOE can identify those important interactions that might be neglected if the One Factor at a Time (OFAT) experimental method is employed. DOE works together with RSM, where RSM is a technique used to generate a mathematical model that relates the design parameters and the output variable [22]. Up to the present time, bimetallic copper-zirconia with imidazolate framework materials have not been studied for photocatalytic reduction of $CO_2$ to methanol in a liquid medium.

## 2. Results

### 2.1. Phase Analysis of Catalyst Components

Phase analysis of catalyst components was investigated by XRD technique. Figure 1 displays the diffraction patterns of imidazolate based CuZrIm catalysts with different Zr loadings. For comparison, an XRD spectrum of CuIm was also included. One prominent peak was detected for each sample at a 2θ value of 30.5° indicating the orthorhombic structure of Cu-N bond angle (ICDD card No.-01-0768567). Likewise, diffraction pattern with peaks at 32°, 35°, 37° and 39.8° on the 2θ scale was found which is indexed as tetragonal phased Cu-N bond angle with ICDD card No.-01-0803170 in CuIm catalyst.

The distribution of different phases was affected by the different loadings of zirconia on CuIm. A catalyst with lower zirconia loading (0.5 mmol) was crystalline. A well-dispersed and highly amorphous phase of Cu (II) ion indicated due to lower crystallinity in the sample. Furthermore, no crystalline peak was observed for CuZrIm1 and CuZrIm1.5, suggesting that CuZrIm1 and CuZrIm1.5 are in the amorphous phase which could not be detected by XRD. As seen from the results of XRD patterns, the Cu-N bond angle at 30.5° reflection width became broader and weaker than the corresponding CuIm in higher loading of zirconia materials. It indicates that hybrid photocatalyst has lower crystallite and smaller particle size.

The diffraction peak at 45.5° shows the presence of zirconia (Zr). In fact, the increase in Zr content is accompanied by a decrease in intensity and a slight increase in the width of the Zr (IV) diffraction peak. This observation indicates that there are a reduction in the Zr (IV) crystallization degree and an improvement in the zirconia dispersion rate. Nevertheless, as the zirconia loading is 0.5 mmol, the intensification in Zr (IV) diffraction peak is apparent,

indicating that there is a rise in the crystallization degree and a decrease in the zirconia dispersion rate. No separate diffraction peaks were observed for the Zr phase in the catalysts, however, for Zr loadings of 1 mmol and 1.5 mmol. Hence, the zirconia is well-dispersed (amorphous) under these loading conditions. As observed from the results of XRD patterns, with an increase of metal loading, the oxide phase produce which reveals that the crystallinity of bond angle declines and the crystallite size becomes larger [12]. From the XRD results, it can be deduced that by employing the optimum amount of CuZrIm1 catalyst, the inclusion of Zr can promote the dispersion and reduce the crystallization for both active components (Cu and Zr). As noted earlier, the highly dispersed form of active components could lead to better catalyst performance. From the XRD findings, it seems that CuZrIm1 is the most active catalyst among the synthesized catalysts.

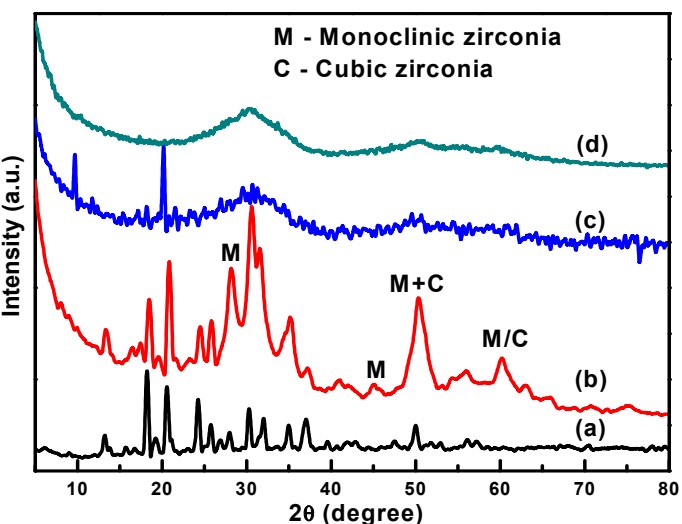

**Figure 1.** XRD pattern of (**a**) CuIm (**b**) CuZrIm0.5 (**c**) CuZrIm1 (**d**) CuZrIm1.5.

The chemical states of catalyst components were evaluated by XPS. Figure 2 shows the XPS spectra of Cu 2p of CuIm and CuZrIm materials with different loadings of zirconia. As observed, the CuIm catalyst exhibits the Cu $2p_{3/2}$ parental peak at 934.5 eV. This parental peak is followed by a broad shake-up peak at ~944 eV, thus confirming the existence of $Cu^{2+}$ in the sample. Meanwhile, the additional parental peak of Cu $2p_{1/2}$ occurred at 953.8 eV and satellite peak at ~962.8 eV [23].

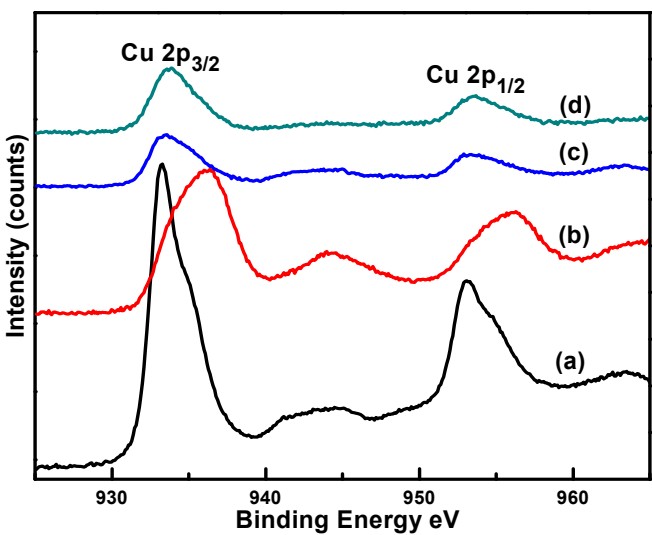

**Figure 2.** XPS Cu 2p spectra of (**a**) CuIm (**b**) CuZrIm0.5 (**c**) CuZrIm1 (**d**) CuZrIm1.5.

Upon adding Zr into CuIm, each catalyst shows core electron peaks at binding energies of ~934 eV and ~954 eV (or Cu $2p_{3/2}$ and Cu $2p_{1/2}$ parental peaks, respectively). Each core electron peak is associated with a satellite peak of binding energy gap 9 eV at 943 eV and 962 eV [19]. The association of shake-up peaks with the parental peaks shows that Cu is found mainly in the form of $Cu^{2+}$ in the CuZrIm catalyst.

The Zr 3d XPS spectra are shown in Figure 3. Two apparent XPS peaks which correspond to Zr $3d_{5/2}$, as well as Zr $3d_{3/2}$, are detected at 182.2 eV and 184.8 eV, respectively. The occurrence of two XPS peaks with an energy gap of 2.6 eV indicated the presence of $Zr^{4+}$ as $ZrO_2$ [24–27]. Additionally, N 1s spectra of the synthesized catalyst can be seen in Figure 4. The binding energies of N 1s spectra in all samples at around between 399–401 eV, which shows the presence of N bond angle with metal atoms.

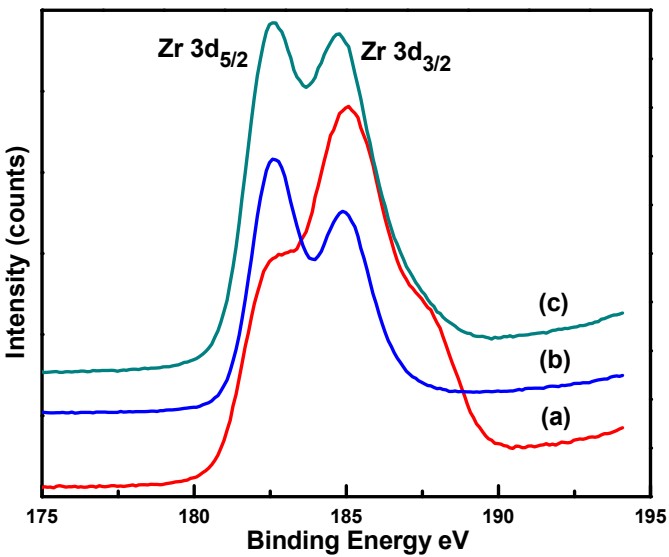

**Figure 3.** XPS Zr 3d spectra of (**a**) CuZrIm0.5 (**b**) CuZrIm1 (**c**) CuZrIm1.5.

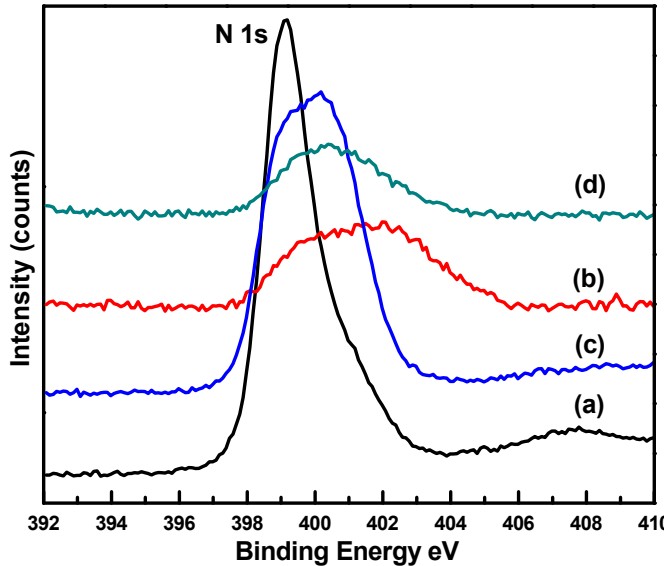

**Figure 4.** XPS N 1s spectra of (**a**) CuIm (**b**) CuZrIm0.5 (**c**) CuZrIm1 (**d**) CuZrIm1.5.

Table 1 reports the full width at half maximum (FWHM) values of binding energies of Cu $2p_{3/2}$ and Zr $3d_{5/2}$. As shown, the increase in Zr content from 0.5 mmol to 1 mmol decreases the binding energy of Cu $2p_{3/2}$ from 936.4 eV to 934.3 eV. The present

observation shows that the interaction between the Cu and the imidazolate framework is now more apparent. Furthermore, there is an enhancement in the copper dispersion rate. By increasing the Zr content, the Cu $2p_{3/2}$ peak position shifts to higher binding energy level. Nevertheless, upon including Zr into the parent catalyst (CuIm), both metal-metal interaction and copper dispersion rate are negatively affected.

**Table 1.** XPS data of CuIm and CuZrIm catalysts with different Zr loadings.

| Catalyst | Binding Energies (eV) | | FWHM (eV) | | Binding Energies (eV) N 1s |
|---|---|---|---|---|---|
| | Cu $2p_{3/2}$ | Zr $3d_{5/2}$ | Cu $2p_{3/2}$ | Zr $3d_{5/2}$ | |
| CuIm | 933.2 | - | 4.22 | - | 399.3 |
| CuZrIm0.5 | 936.4 | 185 | 4.39 | 5.51 | 401.5 |
| CuZrIm1 | 934.3 | 182.6 | 4.17 | 4.36 | 400.2 |
| CuZrIm1.5 | 934.6 | 182.7 | 4.28 | 4.40 | 400.5 |

On the other hand, the binding energies of the Zr $3d_{5/2}$ core electron levels agree well with those published for $Zr^{4+}$ cations [28]. In fact, the peak maxima of Zr $3d_{5/2}$ on the binding energy scale is dependent on the Zr content. The increase in Zr content from 0.5 mmol to 1 mmol can shift the peak from higher to lower binding energy level. This shows that the interaction between the Zr and the imidazolate framework is significant. The binding energy of Zr $3d_{5/2}$ increases when the Zr content increases to 1.5 mmol. The binding energy of Zr $3d_{5/2}$ is the highest (i.e., 185 eV) when the Zr content in the catalyst is the lowest (i.e., 0.5 mmol). This observation tallies with that reported by Damyanova et al. [29] for mixed metal oxide catalyst with varying Zr content. The similar observation can be seen in N 1s spectra. The addition of Zr content into CuIm shifts the peak from 399.3 eV to 401.5 eV.

In addition, the $Cu^{2+}$ ion is surrounded by nitrogen atoms and it has a nearly square-planar symmetry. Therefore, if the symmetry of $Cu^{2+}$ ion is distorted, the charge-transfer excitation would occur due to the electron correlation effect. Furthermore, the broad shake-up in the satellite peak of the Cu 2p profile of the CuZrIm catalyst would happen. In order to further examine this condition, the relation between the FWHM value of the parent Cu $2p_{3/2}$ peaks and the Zr content was studied. Initially, the FWHM value decreases as the Zr loading increases from 0.5 mmol to 1 mmol. However, the FWHM value increases slightly with a further addition of Zr content. The FWHM value is the largest when the Zr content is 0.5 mmol. Hence, the addition of Zr can distort the symmetry of $Cu^{2+}$ ion significantly. Initially, the FWHM value of Zr $3d_{5/2}$ decreases with respect to the Zr content (i.e., from 0.5 mmol to 1 mmol). However, the change in FWHM is insignificant with a further increase in Zr content.

The PL signals of copper-based imidazolate framework catalysts with different Zr loadings are shown in Figure 5. The emission spectra of PL signals show the separation and recombination of electron-hole pairs. For photocatalysts, the fluorescence intensity is related to the efficiency of electron-hole pair recombination [30]. The initial excitation wavelengths of all catalysts were set to 375 nm. When the emission intensity increases, it can be seen that the recombination efficiency of the photo-generated electron-hole pairs increases as well. The intensity of the emission band of CuZrIm1.5 is the highest. Therefore, the recombination efficiency of the photo-generated electron-hole pairs in CuZrIm1.5 increases significantly. The PL emission intensity curve of CuZrIm1 is sharper and broader than those of CuZrIm0.5 and CuIm at 525 nm due to the fact that the photo-generated charges are transferred rapidly from the organic ligand to the Cu-N cluster in CuZrIm1. As a result, the recombination efficiency of photo-induced electron-hole pairs is lesser in CuZrIm1. Photoluminescence (PL) spectroscopy shows that the bimetallic copper-zirconia based imidazolate framework catalyst exhibits better charge-transfer properties.

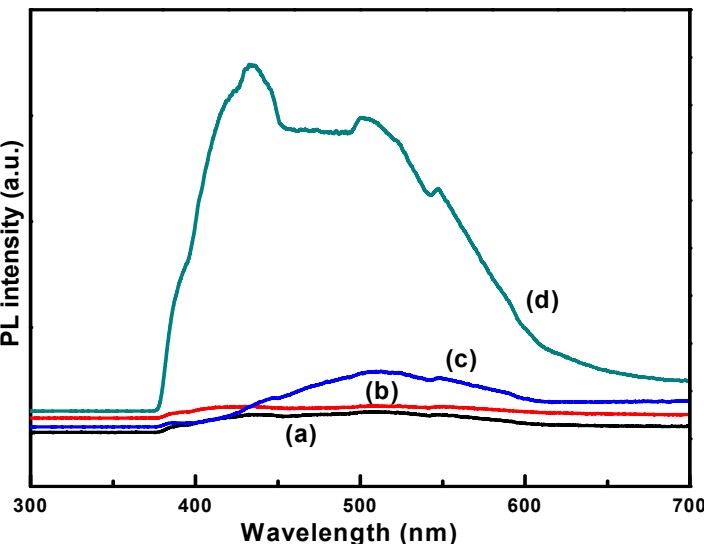

**Figure 5.** PL spectra of (**a**) CuIm (**b**) CuZrIm0.5 (**c**) CuZrIm1 (**d**) CuZrIm1.5.

## 2.2. Photocatalytic Activity

CuIm and CuZrIm catalysts were tested for $CO_2$ photoreduction in liquid media and activity bar graph can be seen in Figure 6. All synthesized catalysts produce the methanol under visible light illumination. The methanol synthesis rate was affected by the different loading of zirconia content in the CuIm catalyst. Nevertheless, a decrease in methanol production rate was observed for catalysts with Zr loading (0.5 mmol) into the parent catalyst. However, the rate of methanol synthesis increased on further addition of Zr content. Consequently, the photocatalytic methanol synthesis rate activity is enhanced in CuZrIm1 and then decreased with further increment in Zr content. Thus, the photocatalytic reduction of $CO_2$ in the system substantially affected.

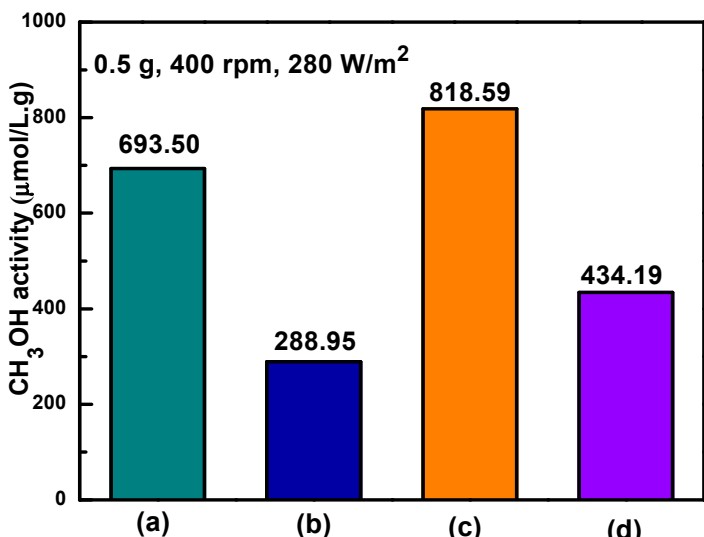

**Figure 6.** Photocatalytic activity histogram of (**a**) CuIm (**b**) CuZrIm0.5 (**c**) CuZrIm1 (**d**) CuZrIm1.5.

The inclusion of bimetallic Cu-Zr on imidazolate framework promotes the photocatalytic activity of CuZrIm which is beneficial in separating the photo-generated electron-hole pairs. However, beyond the critical Zr mass ratio (i.e., 1.5 mmol) in bimetallic photocatalyst, a decay in the photocatalytic activity is apparent. This could be due to the decreasing number of available surfaces as the Zr content increases. Therefore, 1 mmol of Zr mass

ratio is enough for synthesizing CuZrIm catalyst because the $CO_2$ molecules can be easily adsorbed. However, an excessive amount of Zr would block the active sites of CuZrIm. As a result, the number of available surfaces of the catalyst interacting with the $CO_2$ molecules is lesser. Moreover, a high mass ratio of Zr leads to light scattering and blocks absorption of visible light by CuIm photocatalysts and shows less photocatalytic activity in the production of methanol [31].

### 2.3. Optimization Study

The research work discussed in this part of the paper primarily deals with the response of methanol yield influenced by various process variables. The main objective of the research in this section is to optimize the methanol yield along with the stirring rate, the intensity of light and catalyst loading. RSM is usually performed upon screening the previous design to determine the influential experimental variables. A system response might be affected by numerous variables; therefore, it is practically impossible to consider all these variables during the analysis. In other words, only those significant variables should be chosen [32].

In this work, the Box-Behnken design tool of RSM is employed for selected three factors. Fair conditions for methanol synthesis were stirring rate, the intensity of light and catalyst loading. Before application RSM, these conditions were determined by changing one variable at a time while keeping other variables constant. Table 2 summarizes the essential variables that produce significant effects on methanol yield. All these factors are selected by experiments.

**Table 2.** Levels and variables for the Box-Behnken design.

| Variable | Symbol | Unit | Ranges and Levels | | |
|---|---|---|---|---|---|
| | | | −1 | 0 | +1 |
| Intensity of light | A | W/m$^2$ | 200 | 270 | 340 |
| Stirring rate | B | rpm | 400 | 550 | 700 |
| Catalyst loading | C | g/L | 1 | 1.5 | 2 |

The original set of experiments performed in 17 runs which are recorded in Table 3.

**Table 3.** Experimental runs during optimization.

| Runs | Intensity of Light (W/m$^2$) | Stirring Rate (rpm) | Catalyst Loading (g/L) | Methanol Activity (μmol/L.g) |
|---|---|---|---|---|
| 1. | 270 | 550 | 1.5 | 1053.04 |
| 2. | 270 | 550 | 1.5 | 1033.95 |
| 3. | 270 | 550 | 1.5 | 1034 |
| 4. | 270 | 700 | 2 | 885.14 |
| 5. | 200 | 550 | 1 | 854.13 |
| 6. | 340 | 550 | 1 | 845.65 |
| 7. | 200 | 700 | 1.5 | 648.35 |
| 8. | 200 | 550 | 2 | 679.47 |
| 9. | 200 | 400 | 1.5 | 772.84 |
| 10. | 340 | 550 | 2 | 898.32 |
| 11. | 340 | 400 | 1.5 | 771.28 |
| 12. | 340 | 700 | 1.5 | 818.15 |
| 13. | 270 | 550 | 1.5 | 1054.1 |
| 14. | 270 | 400 | 2 | 825.59 |
| 15. | 270 | 550 | 1.5 | 1054 |
| 16. | 270 | 700 | 1 | 902.43 |
| 17. | 270 | 400 | 1 | 948.28 |

Run 13 gave the maximum methanol yield. The RSM analysis was used to develop a model equation for the methanol yield as written in Equation (1):

$$Methanol\ yield = 1045.82 + 47.33A - 7.99B - 32.75C + 42.84AB + 56.83AC + 26.35BC - 182.07A^2 - 111.10B^2 - 44.36C^2 \quad (1)$$

The methanol yield can be calculated by utilizing the quadratic model Equation (1) for the variables *A* as an intensity of light, *B* as a stirring rate and *C* as a catalyst loading within the prescribed range.

The fitting model suggested by the analysis has been described in Table 4. The ANOVA results were analyzed for the validation of the responses to the variables as discussed in the previous section.

**Table 4.** Fitting model for the validation.

| Source | Sequential $p$-Value | Lack of Fit $p$-Value | Adjusted R-Squared | Predicted R-Squared |
|---|---|---|---|---|
| Linear | 0.6987 | <0.0001 | −0.1070 | −0.4279 |
| 2FI | 0.7890 | <0.0001 | −0.3017 | −1.2893 |
| Quadratic | <0.0001 | 0.1186 | 0.9849 | 0.9195 |
| Cubic | 0.1186 | | 0.9930 | |

The analysis of variance (ANOVA) analysis reported in Table 5 can be used to determine the significance levels of the linear, quadratic and interaction terms in the model via examining the probability value ($p$-value). The term is significant if the $p$-value is less than 0.005 [32]. The proposed model should provide an adequate approximation to the real system. From the presented ANOVA results, almost all independent and interaction terms are significant.

**Table 5.** Analysis of the ANOVA model.

| Factor | F-Value | $p$-Value |
|---|---|---|
| Model | 116.99 | <0.0001 |
| A-Intensity of light | 70.75 | <0.0001 |
| B-Stirring rate | 2.02 | 0.1986 |
| C-Catalyst loading | 33.87 | 0.0007 |
| AB | 28.98 | 0.0010 |
| AC | 51.01 | 0.0002 |
| BC | 10.97 | 0.0129 |
| $A^2$ | 551.05 | <0.0001 |
| $B^2$ | 205.19 | <0.0001 |
| $C^2$ | 32.71 | 0.0007 |

ANOVA is a statistical procedure that can be employed to check the significance and adequacy of a model [33]. From the ANOVA results, the correlation coefficient of the regression model representing the photocatalytic production of methanol is high ($R^2$ = 0.99). As the correlation coefficient is close to one, the goodness of fit between the model and the experimental data is promising [34]. The adjusted $R^2$ could be used to measure the goodness of fit between the model and the experimental data if the comparison between models of different numbers of the independent variable is made.

The F value is the ratio between the mean square of the model and the residual error. It indicates the significance level of each controlling factor on the tested model [35]. From Table 6, the F value of the model is 116.99 and the $p$-value is <0.0001. Therefore, the model is statistically significant ($p < 0.005$).

**Table 6.** Optimized reaction condition generated by DOE software.

| Parameters | Intensity of Light (W/m$^2$) | Stirring Rate (rpm) | Catalyst Loading (g/L) |
|---|---|---|---|
| Optimum conditions | 275 | 540 | 1.3 |

The accuracy of the model can be checked by evaluating the R$^2$ and the adjusted R$^2$ values. These values are 0.9934 and 0.9849, respectively. The predicted R$^2$ is related to the fitting error of a model. In this case, the predicted R$^2$ is 0.9195, which is comparable to the adjusted R$^2$ of 0.9849. Analysis of variance (Table 5) also showed that the regression model for methanol production is statistically good with a significance level of $p < 0$. Thus, well-fitting models for methanol production are successfully established.

Adequate precision enables us to measure the signal to noise ratio. It also compares the range of the predicted values at the design points to the average prediction error. If the signal to noise ratio is greater than 4 then it is desirable and indicates an adequate signal. In the present work, the signal to noise ratio is found to be 32.059. This value of rate confirms the reliability of the experiment data. The degree of precision is determined by the coefficient of variation (C.V.% = 5.50) and standard deviation (SD = 3.05). The adequacy with which experiments were conducted confirmed by low values of CV and SD. The models have high R$^2$ value, significant F-value, low standard deviation, and coefficient of variance. High precision in predicting methanol response is indicated by these results. Therefore, the model was used for further analysis.

Figures 7–9 show the visualized effects of the experimental factors on the conversion response. Figure 7 provides the graphical way of validating the model, which shows the points are approximately lying on the straight line which indicates that the set of data are normally distributed. The normality is linear; therefore, response transformations is not needed for the model.

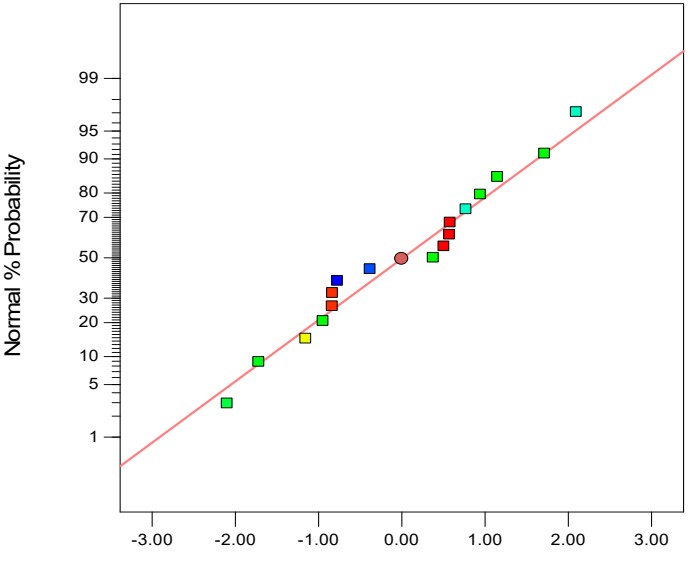

**Figure 7.** Normal percentage probability plot.

The actual and predicted values are shown in Figure 8. The actual data represents the measured response data of a run, and the predicted data is generated from the mathematical model. From Figure 8, the R$^2$ and adjusted R$^2$ values are found to be 0.9934 and 0.9849, respectively. The prediction is good as the data fall along the 45° line. The drift in the model was tested using the Run order plots as shown in Figure 9. The Run order plot is a scatter

plot containing all residuals plotted against an index to represent the order of the composed data. It is particularly useful for analyzing predictor variables of randomized run order. For the present case, the uniform spread of the residuals across the range indicates that the model has no apparent drift.

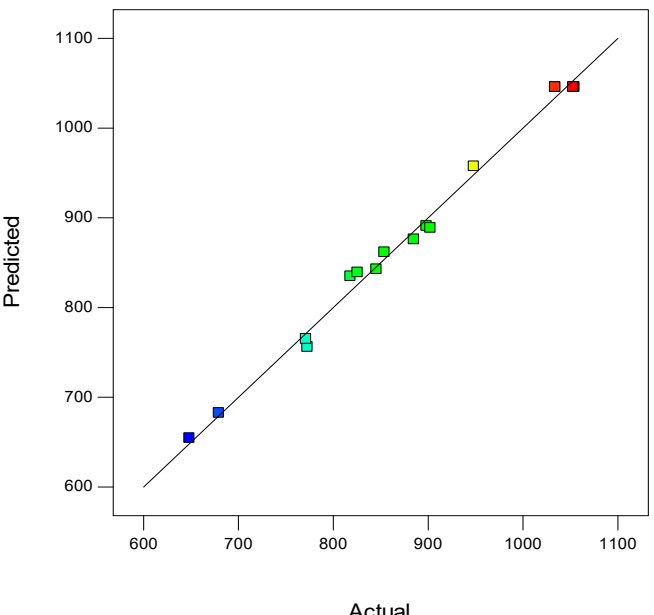

**Figure 8.** The actual versus predicted plot.

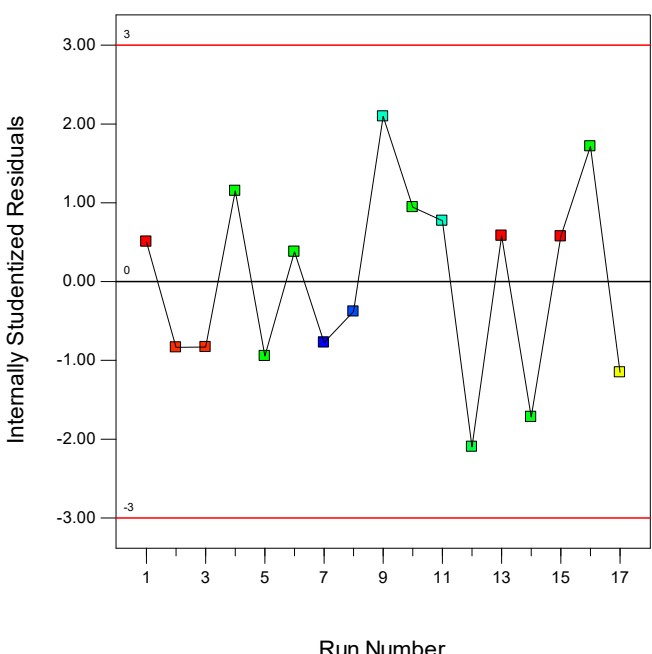

**Figure 9.** Residual versus run order plot.

### 2.4. Effect of Process Parameters on Methanol Yield

To investigate the effect of the three factors on the conversion of methanol, the RSM was used, and three-dimensional (3D) plots and two-dimensional (2D) contour plots were drawn. Based on the acquired ANOVA results- stirring rate, the intensity of light and catalyst loading were found to have significant effects on the yield of methanol. The 3D

response surface diagram and 2D contour plots for all variables are shown in Figures 10–15. These figures are useful in identifying the area where the methanol yield can be optimized.

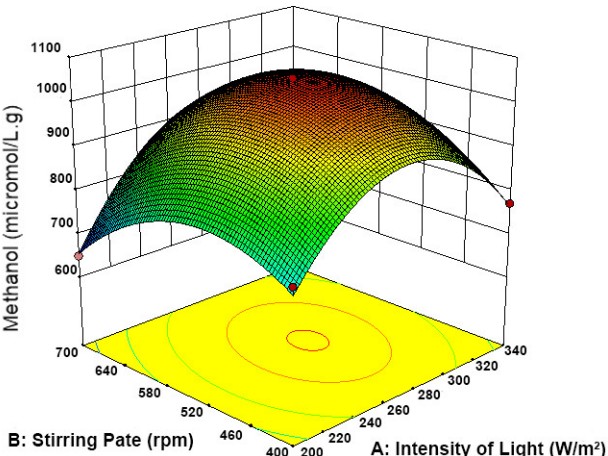

**Figure 10.** Effect of intensity of light and stirring rate on methanol yield.

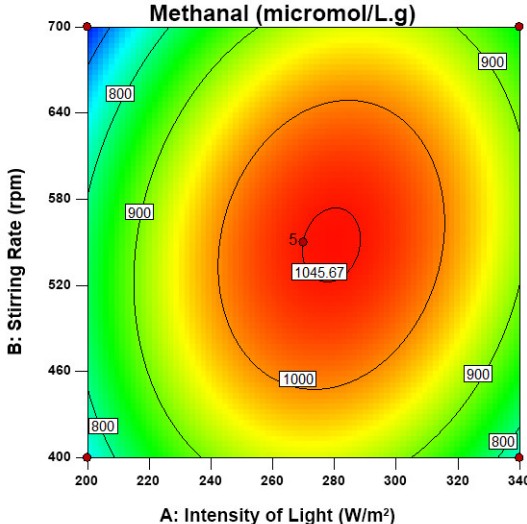

**Figure 11.** 2D contour diagram for representing methanol yield based on intensity of light and stirring rate.

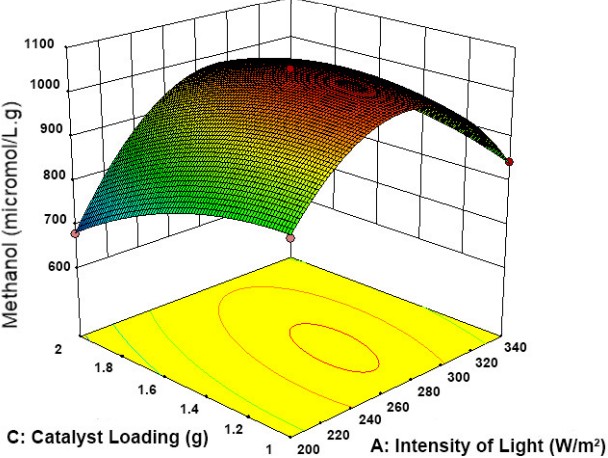

**Figure 12.** Effect of the intensity of light and catalyst loading on methanol yield.

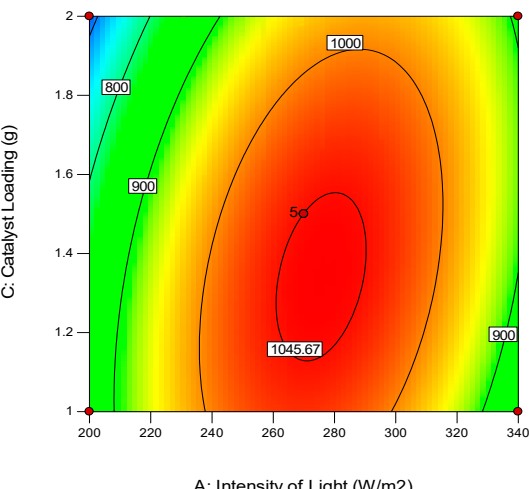

**Figure 13.** 2D contour diagram for representing methanol yield based on intensity of light and catalyst loading.

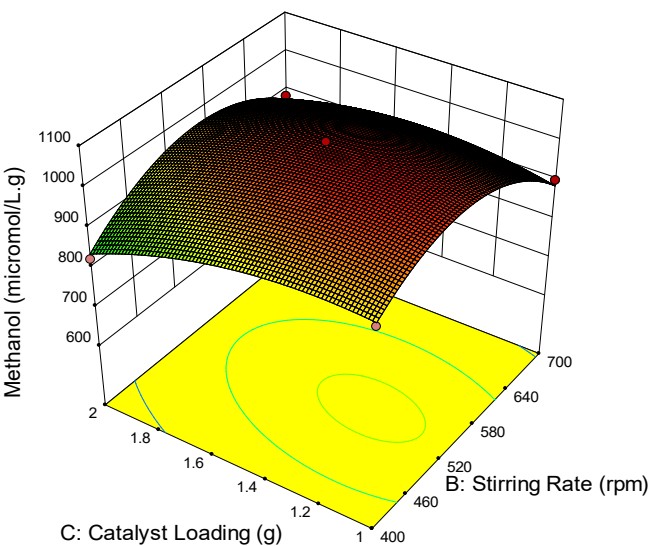

**Figure 14.** Effect of stirring rate and catalyst loading on methanol yield.

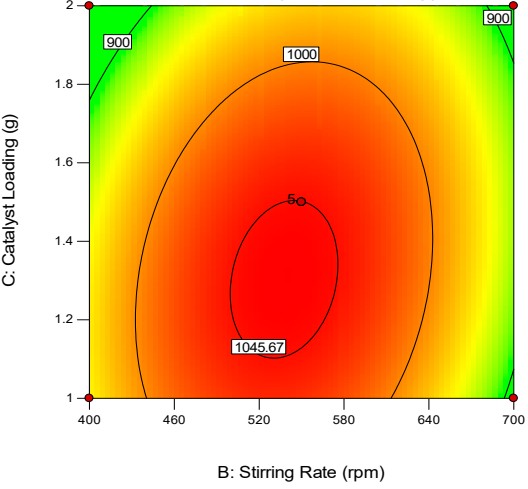

**Figure 15.** 2D contour diagram for representing methanol yield based on stirring rate and catalyst loading.

From Figure 10, it can be observed that as the methanol yield increased with the increase of stirring rate and intensity of light up to the 550 rpm and 280 W/m$^2$, respectively.

It is observed that maximum methanol yield could be attained if the stirring speed and the light intensity for photocatalysts are optimal. The difference in methanol yield is attributed to the change in thickness of the boundary layer surrounding the catalyst particles. The boundary layer thickness changes with respect to the stirring rate. As stirring rate increases, boundary layer thickness would decrease to ~550 rpm. Therefore, product transport rate increased from catalyst surface to bulk liquid volume observed enhanced methanol production. Apparently, increased stirring rate from 550 rpm to 700 rpm would decrease the methanol yield. This observation shows that the liquid methanol production from $CO_2$ is a highly diffusive and mass transfer-controlled process. Decreased stirring rate (400 rpm) leads to smaller mass transfer rate inside the pores and voids of catalyst (or lower methanol yield [36]). The yield value plateaus at a certain level at increasing light intensity. Note, the yield depends on other factors such as irradiation time, reaction medium, catalyst active sites, and reactor configuration as well [37–39]. Figure 11 shows the 2D contour diagram of two variables, i.e., stirring rate and light intensity.

From the 3D surface diagram in Figure 12 describes that the methanol yield increases with the increase of catalyst loading. The catalyst loading has more dominant over the intensity of light in terms of increasing the methanol yield. Noticeably, a decrease in methanol yield upon increasing the photocatalyst loading (2 g/L) in the reaction chamber can be attributed to the mass transfer controlled process [40], reduced solar light exposure to photocatalyst [41] and formation of boundary layers around the photocatalyst particles obstructing reactant to reach the reaction sites [42]. Figure 13 presents the 2D contour diagram of methanol yield in terms of two variables catalyst loading and intensity of light.

According to Figures 14 and 15, the methanol production improved with the increase in catalyst loading as well as in stirring rate. Methanol production is increased with increasing the loading of the photocatalyst in the reactor chamber while stirring rate effects on the methanol yield up to the 550 rpm. Above 550 rpm stirring rate, the slurry would become very violent, and the liquid will start to splash on the walls above the liquid surface and resulted in the loss of catalyst and consequently lowering the reliability and accuracy of the acquired results. The ranges of the two variables for the number of intervals can be visualized in the 2D contour diagram which is shown in Figure 15.

## 2.5. Optimum Reaction Conditions

Responses such as methanol production yield and running cost can be optimized using the Box-Behnken DOE model. Single-objective optimization is not recommended for the present case, as changing one factor can lead to different effects on the response variables. In this research work, multi-objective optimization with the desirability of 1 is performed. The desirability function is one of the most significant multi-criteria aspects of mathematics [22]. This methodology involves the construction of a desirability function for each response. The mathematical model developed in this study was applied to find the optimum and significant process parameters in order to get the maximum methanol yield. The main objective is to maximize the photocatalytic production efficiency upon attaining the optimum values of reaction parameters. These optimal reaction parameters were summarized in Table 6. The experimental design strategy can be adapted to determine the optimum values of reaction parameters for maximizing the photocatalytic production rate. The optimization results of one of the catalysts were depicted in the ramp view diagram shown in Figure 16.

After the optimization process, a validation experimental study has been carried out with the optimized conditions and the result found a maximum activity of methanol of ~1054 μmol/L.g which is maximum yield in the tests conducted.

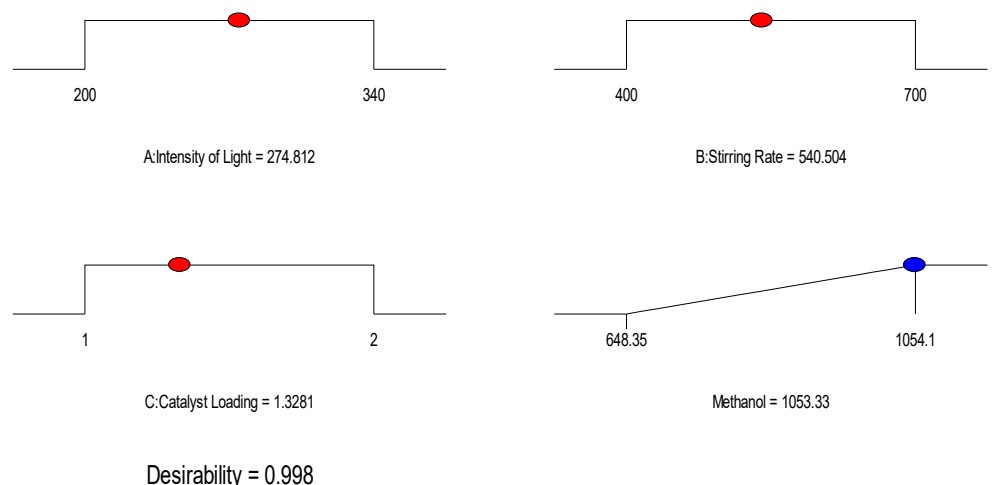

**Figure 16.** Ramp view diagram for optimization.

## 3. Discussion

Due to the different catalyst preparation method, reaction conditions and reactor setup, an exact comparison cannot be made of the present study with previous works, but a general statement can be made that the production of methanol can be increased with the addition of dual metal into imidazolate framework. The photocatalytic activities of synthesized materials in this study for reducing $CO_2$ to $CH_3OH$ were evaluated in a continuous-flow reactor system under visible light irradiation (400–800 nm). The results indicate that the sample CuZrIm1 (bimetallic mixed imidazolate) exhibits the best photocatalytic performance. The methanol yield over the sample CuZrIm1 is also higher than that of Cu-porphyrin/MOF (5.97 μmol/L.g) [21], ZIF-8/$Zn_2GeO_4$ (2.44 μmol/L.g) [9] and Ti-MCM-48(25)CuTPP (297.06 μmol/L.g) [1] under the same reaction conditions. The main reason could be attributed to the smallest fluorescence intensity of CuZrIm1 with the presence of bimetallic active sites for $CO_2$ adsorbance, which gives the best performance toward visible light.

However, the methanol yield of CuIm (1712.7 μmol/L.g) [12] is better than that of CuZrIm1 (1054 μmol/L.g). However, the yield of methanol is expensive using CuIm catalyst due to the usage of a high amount of catalyst precursors, and a high intensity of light compared to the prepared CuZrIm1. Besides this, the main reason attributed to the reactor size (200 mL jar capacity), which decreases mass transfer limitation. The yield of methanol using synthesized catalyst CuZrIm1 is compared with the literature data as tabulated in Table 7.

**Table 7.** Recent progress in photocatalytically reduction of $CO_2$.

| Sr. No. | Catalyst | Reactant | UV Cut-off Filter Wavelength (nm) | Xenon Lamp | Methanol (μmol/L.g) | Year |
|---|---|---|---|---|---|---|
| 1. | Cu porphyrin-based MOF | $CO_2$ and a solution of the triethylamine | $\lambda \geq 420$ | 300 W | 5.97 | 2013 |
| 2. | ZIF-8/$Zn_2GeO_4$ | $CO_2$ and a solution of sodium sulfite | No filter used | 500 W | 2.44 | 2013 |
| 3. | CuIm | $CO_2$ and a solution of sodium hydroxide and sodium sulfite | $\lambda > 400$ | 500 W | 1712.7 | 2013 |
| 4. | Ti-MCM-48(25)CuTPP | $CO_2$ and a solution of sodium hydroxide and sodium sulfite | No filter used | 500 W | 297.06 | 2018 |
| 5. | CuZrIm1 | $CO_2$ and a solution of sodium hydroxide and sodium sulfite | No filter used | 500 W | 1054.0 | Present study |

## 4. Materials and Methods

Materials such as zirconium sulphate pentahydrate ($Zr(SO_4)_2 \cdot 5H_2O$), imidazole (Im), copper sulphate pentahydrate ($CuSO_4 \cdot 5H_2O$) were purchased from R & M Chemicals (Subang Jaya, Selangor, Malaysia) while sodium hydroxide (NaOH) and sodium sulfite ($Na_2SO_3$) were obtained from Sigma Aldrich (Subang Jaya, Selangor, Malaysia) and ammonia ($NH_3$) was purchased from Merck (Petaling Jaya, Selangor, Malaysia). Deionized (DI) water was used in the synthesis of all catalysts and reaction medium. No further purification of all reagents was performed and used directly.

### 4.1. Synthesis of CuZrIm Catalyst

Imidazole was used as support without any purification after received. The powder chemicals were mixed separately in DI water, and then the mixture was stirred for ten minutes. The copper salt solution (2 mmol) was mixed with the ammonium hydroxide solution (4 M) before mixing with imidazole (4 mmol) solution to prepare CuIm catalyst. To synthesize CuZrIm, the solution of copper sulphate (2 mmol) dropwise poured into ammonium hydroxide solution (4 M) and then copper and zirconia (0.5, 1, and 1.5 mmol separately) solutions were mixed dropwise into an imidazole (4 mmol) solution under continuous stirring at room temperature. The mixture was poured into Teflon-lined stainless-steel autoclave and then hydrothermally treated at 110 °C for two days. The solid product was obtained by filtration and washed with DI water for 3–4 times. The prepared catalyst was finally dried at 85 °C overnight. The method adopted for the synthesis of catalysts was taken from Li et al. [12]. The schematic representation of catalyst synthesis is presented in Figure A1.

### 4.2. Photocatalytic Study

Catalytic activity and optimization study of catalysts were tested in an optical fiber photocatalytic reactor. The detailed set up of the photocatalytic study has been reported in previous work [43]. The schematic representation of photocatalytic reduction is shown in Figure A2.

A screening study of catalysts (each 0.5 g) was carried out in 500 mL solution of sodium hydroxide and sodium sulfite with a 400 rpm (revolutions per minute) stirring speed. The liquid samples were collected every hour of the interval through a glass syringe and then centrifuged for 10 min at 6000 rpm for the settling down of solid catalyst particles. To confirm the photoreduction of $CO_2$, blank tests were conducted. The blank tests were carried out with the catalyst in the dark and without the catalyst in the solar light. Pure $CO_2$ was used as a reacting gas. Prior to reaction study, the reactor was purged with the reactant gas to eliminate any liquified air into the reactor solution. Finally, the 0.5 g of catalyst was loaded to the reaction solution from the top of the reactor by holding the inner part and then fixed it. The reactant gas was transferred to a reactor with stainless steel tubing. Before the lamp was turned on, the reaction solution was allowed to stir further for 30 min to absorb the $CO_2$ over the catalyst surface.

### 4.3. Optimization Study

In this research, the RSM technique of DOE is applied to carry out the experimental works and to determine the cause and effect relationships of the variables and responses. The molar ratios of zirconia were used to synthesize different bimetallic Cu-Zr based imidazolate framework photocatalysts for methanol production. To clearly understand the maximum methanol production, three variables were tested. Table 8 presented the three variables which were used in the optimization study.

**Table 8.** Three different variables used in the optimization.

| Process | Variables |
|---|---|
| Intensity of light (W/m$^2$) | 200, 270, 340 |
| Stirring rate (rpm) | 400, 550, 700 |
| Catalyst loading (g/L) | 1, 1.5, 2 |

In the Design-Expert software (Tieto Sdn Bhd, Kualalumpur, Malaysia), each significant independent variable is examined at three levels (−1, 0, 1) [32]. The methanol yield results obtained from the experiment are then analyzed using various regression to fit in Equation (2):

$$Y = \varphi_0 + \sum_{i=1}^{n} \varphi_i x_i + \sum_{i=1}^{n} \varphi_{ii} x_i^2 + \sum_{i=1}^{n-1} \sum_{j=i+1}^{n} \varphi_{ij} x_i x_j \tag{2}$$

where, $Y, \varphi_0, \varphi_i, \varphi_{ii}$ and $\varphi_{ij}$ indicate the predicted response, the intercept term, the linear coefficient, the squared coefficient, and the interaction coefficient, respectively.

*4.4. Characterization of CuZrIm Catalyst*

By using the K-Alpha X-ray Photoelectron Spectroscopy (XPS) system (D8-Advance, Bruker, Billerica, MA, USA, CuKα radiation, λ = 1.54 Å), the surface compositional charges were analyzed by applying a monochromatic AlKα source (hν = 1486.6 eV, 150 W). The crystal structures of the catalysts and lattice parameters at room temperature were identified using XRD with a scanning rate of 2°/min to cover the range of angles: 10° < θ < 80°. The Scherrer equation (Equation (3)) was then used to compute the crystal size:

$$D = K\lambda / \beta Cos\,\theta \tag{3}$$

where $D$ is the crystallite size, $K$ is the crystallite shape factor (0.9), $\lambda$ is the wavelength (=1.54 nm), $\beta$ is the full width at half of the maximum intensity of the peak (in radian), and $Cos\,\theta$ is the angle (position) of the peak at the position of 2θ.

The photoluminescence spectrum of the synthesized photocatalyst materials was obtained to apprehend the recombination of electron-hole, the transfer of charge and the emission state of the photocatalyst material. As such, an LS 55 luminescence spectrometer (Perkin Elmer, Billerica, MA, USA) with a pulsed xenon lamp was employed as the excitation source.

**5. Conclusions**

In this work, bimetallic copper-zirconia based imidazolate framework photocatalysts were developed for investigating $CO_2$ reduction to methanol in an optical reactor. All catalysts in this study were successfully synthesized by hydrothermal method. XRD results and XPS investigations have also confirmed the synthesis of the target molecule and PL spectra shows photocatalyst absorbance in the visible region makes it a valid candidate for photocatalytic application. The activity of the catalyst was enhanced with the addition of Zr to the parent CuIm catalyst, and the best activity was recorded for catalyst with 1 mmol of Zr content. The screened catalyst with the best yield (CuZrIm1) was investigated via RSM for a suitable intensity of light, stirring rate, and photocatalyst loading. Investigation studies for the intensity of light were conducted at different light intensity such as 200, 280, and 340 W/m$^2$, different stirring rate such as 400, 550, and 700 rpm, and various amount of catalyst loadings such as 1, 1.5 and 2 g of catalyst. Variation in the reaction parameters affected the activity profile of catalyst. RSM gives an optimum reaction condition for the catalyst which is given as—275 W/m$^2$ light intensity, 540 rpm stirring rate and 1.3 g of catalyst loading. The highest methanol yield over CuZrIm1 catalyst (1054 µmol/L.g) could be attributed to the presence of uniformly dispersed metal ions and more significant half oxidation (+1.28 V) and reduction (−1.70 V) potential values calculated with the equation reported by Wang et al. [44]. It can be concluded that synthesized photocatalyst has shown

great potential towards visible light, which can convert the greenhouse gas to the valuable liquid product such as methanol. Methanol is considered as an alternative liquid fuel. Furthermore, this study provides a suitable path for effective photocatalytic $CO_2$ reduction to methanol under visible light irradiation.

**Author Contributions:** Resources, G.S.J.; Supervision, C.F.K., B.A. and L.J.W.; Writing—original draft, S.G.; Writing—review & editing, M.S.S. All authors have read and agreed to the published version of the manuscript.

**Funding:** "This research was funded by Universiti Teknologi PETRONAS (YUTP-FRG), 0153AA-E98" and "The APC was funded by YUTP-FRG".

**Acknowledgments:** The financial assistance from the Foundation of Universiti Teknologi PETRONAS (YUTP-FRG) with the cost center 0153AA-E98 is gratefully acknowledged.

**Conflicts of Interest:** The authors declare no conflict of interest.

**Appendix A**

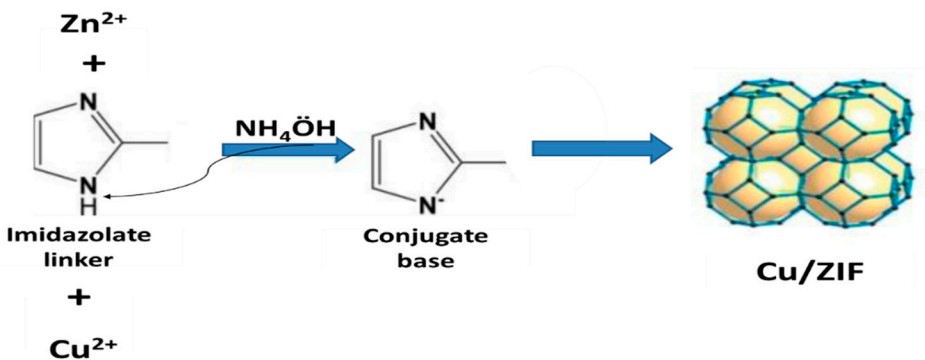

**Figure A1.** Schematic representation for construction of Cu-based ZIF catalyst.

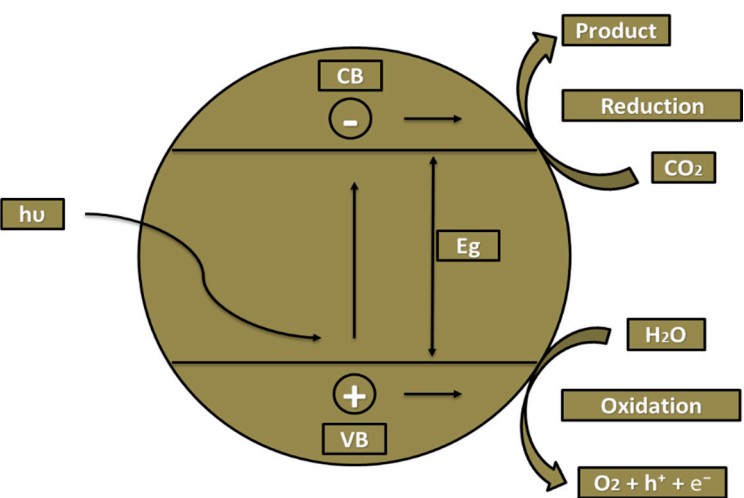

**Figure A2.** Schematic representation of photocatalytic $CO_2$ reduction.

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
