# Peer review of "Photocatalytic Reduction of CO2 to Methanol Using a Copper-Zirconia Imidazolate Framework"

_catalysts, doi:10.3390/catal11030346_

Round 1

Reviewer 1 Report

Goyal et al. reported copper-zirconia imidazolate (CuZrIm) frameworks and it has been used for photocatalytic reduction of CO2 to methanol.

Comments to the authors:

  1. The authors should fit all the XPS graphs e.g. Figures 2, 3, and 4 and assign the peaks.
  2. The authors should provide the SEM and XRD images of CuZrIm frameworks before and after catalytic reduction to confirm the stability of the catalyst.
  3. The authors should provide the scheme for the synthesis and photocatalytic CO2 reduction.
  4. The authors should provide a table for methanol yield comparing with other catalysts.

Reviewer 2 Report

In the present manuscript the photocatalytic reduction of carbon dioxide to methanol with the help of metal-organic frameworks is well described, this is very relevant in light of the problem of global warming (although, judging by the current winter in the Northern Hemisphere, there is no global warming, but rather a global cooling begins). However, seriously, I think the article will be interesting for readers, and I will recommend it for publication. It may not be worth using the mathematical term "graph" in the figure captions, but replace it with the usual terms used in the manuscript: X-ray diffraction pattern, XPS spectra etc. However, this should be left to the discretion of the authors. 

Reviewer 3 Report

The authors describe photocatalytic reduction of CO2 using CuZrIm catalyst and a comparison between various catalytic loading. The article can be published with following edits

  • Reference needs significant improvement for the sections of Introduction and Results.
  • Line 36: band energy gap is generally used in describing the semiconductor or transition metal catalyst. For general organic synthesis, the correct terminology is activation barrier.
  • Line 92-114: Figure 1 depicts that dispersion of Zr in the catalyst increases as Zr loading is increases. However, CuZrIm1 being more active than CuZrIm1.5 is not clear in this section.
  • Figure 2, 3 and 4: CuZrIm1 and CuZrIm1.5 follow similar binding energy peaks as of CuIm. Explain why the peaks are significantly shifted in CuZrIm0.5?
  • English editing throughout the article is recommended.

Round 2

Reviewer 1 Report

The authors should always keep all the raw data with them. The XPS data can be fitted by several softwares like origin, XPS peak fit.

The authors have partially addressed my concerns.